# Use of Stable Isotopes (δ^13^C and δ^15^N) to Infer Post-Breeding Dispersal Strategies in Iberian Populations of the Kentish Plover

**DOI:** 10.3390/ani14081208

**Published:** 2024-04-17

**Authors:** Andrea Gestoso, María Vidal, Jesús Domínguez

**Affiliations:** Department of Zoology, Genetics and Physical Anthropology, Faculty of Biology, University of Santiago de Compostela, 15782 Santiago de Compostela, Spain; andrea.gestoso@rai.usc.es (A.G.); mariajose.vidal@usc.es (M.V.)

**Keywords:** available habitat, *Charadrius alexandrinus*, post-breeding dispersal, stable isotopes

## Abstract

**Simple Summary:**

Beaches are important habitats for migratory birds, but human pressure and sea level rise threaten their availability. This study reveals post-reproductive habitat-use strategies of the Kentish plover (*Charadrius alexandrinus*) in the Iberian Peninsula by analysing C and N isotopes in feathers. Birds from the northwest of the Iberian Peninsula exhibit greater fidelity to a single habitat type, while those from the Mediterranean coast and the Atlantic coast of Andalusia show variety in their dispersal. The lack of alternative habitats, the reduction of beaches due to sea level rise, and human pressure threaten the survival of the species in the northwest of the Iberian Peninsula.

**Abstract:**

Beaches are among the habitats most frequented by migratory birds for breeding and/or wintering. However, threats such as human pressure and sea level rise can reduce the availability of these habitats for different species. The presence of alternative areas, such as salt pans and brackish habitats, is essential for many migratory shorebird populations. This study addresses the post-breeding dispersal of the Kentish plover (*Charadrius alexandrinus*) in the Iberian Peninsula by analysing C and N isotopes in feathers. The study was conducted at six locations along the Iberian coast, which were categorized into three areas: the NW Atlantic coast, the Atlantic coast of Andalusia, and the Mediterranean coast. Although linear mixed models did not reveal any significant effects of sex or coastal area on isotopic levels, the variability in the data suggests different habitat-use strategies in the post-reproductive period. Isotopic levels in birds from the northwest of the Iberian Peninsula exhibit greater fidelity to a single habitat type, while those from the Mediterranean coast and the Atlantic coast of Andalusia show greater variability, indicating different individual dispersal strategies. The lack of alternative habitats for the northwest Iberian population, the reduction in available habitat due to rising sea levels, and human pressure together pose a serious threat to the survival of this species, already with an unfavourable conservation status.

## 1. Introduction

Migratory shorebirds migrate between breeding and staging areas along generally consistent routes [1], frequently using beaches and saline and/or brackish habitats [2,3]. Beaches are important breeding and foraging habitats for shorebirds, but they are threatened by resort development, road construction, coastal erosion, sea level rise, and human disturbance [4,5,6]. Increasing global temperatures will result in increases in sea level due to the expansion of oceanic water and the melting of glaciers and ice sheets [7]. Inundations due to sea level rise could lead to the conversion of intertidal to subtidal habitats and, therefore, a reduction in the availability of habitats to shorebirds [6,8]. Moreover, overwash and the associated consequences are expected to increase because of both sea level rise and the intensification of coastal occupation [9]. The impacts of such habitat changes on shorebird populations will depend on the availability of alternative areas that the birds can use and where similar levels of survivorship and fecundity can be reached [10]. Salt pans and wetlands are included among alternative areas that can support large populations of migratory waterbirds, such as shorebirds [11,12,13,14].

Unravelling migratory connectivity between breeding, stopover, and wintering areas is important to predict the influence of habitat change on population demographics [15,16]. Stable isotope analysis has been used to investigate dispersal and migratory movements in shorebirds [3,16,17,18], as natural patterns of geographic variation have been observed in isotopic ratios on land [19].

The Kentish plover (*Charadrius alexandrinus*) is a common shorebird in the temperate and subtropical belt of Eurasia and N Africa [20]. However, most breeding populations in Europe have undergone a marked decline since the early part of the twentieth century [21].

The Iberian population is the largest in Europe [20] and is concentrated along the coast, being less common in inland wetlands [22,23]. The species has undergone a strong decline in the area it occupies, and it has disappeared from some coastal areas, especially from the Mediterranean coast [23] and also the coast of Portugal [24]. In the northwest Iberian peninsula, the birds breed exclusively on beaches, but along the Portuguese coast, southern Spain and the Mediterranean coast, they also occupy salt pans and salt marshes, as well as sparsely vegetated salt flats and coastal grasslands [2,22,23,25]. Inland populations of Kentish plovers typically nest on the sandy margins of brackish lagoons, but also on the shores of reservoirs or on islands and rice fields [2,22,26,27]. In the context of climate change, the rise in sea level and the increase in the rate or severity of maritime storms are restricting the useful strip of beaches available [28]. This situation increases the importance of supratidal habitats such as salt pans for the species.

The objective of this study was to unravel the post-breeding dispersal strategy in Iberian Kentish plovers through the analysis of the isotopic levels of C and N in the feathers of breeding adults.

## 2. Materials and Methods

### 2.1. Study Area

Samples were collected from six breeding locations situated along the Iberian coast. Among these, three sites are positioned along the Mediterranean coast, while the remaining three are located on the Atlantic coast, with one in southern Spain and two in northern Portugal and northwestern Spain (Figure 1).

The Mediterranean localities include the Ebro Delta wetland, Laguna de la Mata, and Serradal beach (Figure 1). Within the Ebro Delta, recognized as a significant breeding area for the Kentish plover in Spain [29], nests were predominantly observed in zones adjacent to brackish water and coastal beaches. Similarly, nests in Laguna de la Mata were found along areas bordering brackish water, whereas on Serradal beach, nests were situated in deep sandy areas amidst dry interdunal depressions.

The Atlantic location in Andalusia (southern Spain) was the Tinto-Odiel estuary, where breeding areas were located on beaches of the marine zone, while in northern Portugal (Carreço and Ancora beaches) and northwestern Spain (Rostro, Carnota, and Balieiros beaches), nests were found exclusively on coastal sandy beaches (Figure 1).

### 2.2. Methods

Plovers were captured from nests using a funnel-trap and were sexed by dichromatic plumage characteristics [30].

Feathers were collected between March and June 2009 from 44 adult Kentish plovers (17 males and 27 females) breeding on the Mediterranean coast (18 birds), the Atlantic coast of southern Spain (9 birds), and northern Portugal and northwestern Spain (17 birds).

The inner first primary feather of each bird was clipped. Although little is known about temporal and spatial moulting patterns in the Iberian Kentish plover [27], the moulting of primary feathers in European populations usually takes place in the post-breeding period (August-October), although it may begin in June [31,32]. Even so, our samples were mostly obtained between April and May, and only 3 samples were obtained in June. Each feather sample was placed in a (separate) plastic bag and stored at −20 °C prior to analysis.

### 2.3. Isotopic Analysis

The feathers were washed in a solution of 0.25 M sodium hydroxide and pure water to remove waxes and oils. The washed feathers were placed in clean, screw-top vials and dried overnight at 60 °C, in the oven of an elemental analyser. The dried feathers were cut into small sections (<1 mm sections) in the sample vials with surgical scissors.

Stable isotope ratios were measured in a continuous-flow isotope-ratio mass spectrometer (Deltaplus, ThermoFinnigan, SUERC-East Kilbride, UK) coupled to an elemental analyser (FlashEA1112, ThermoFinnigan Instruments, CE Elantech, Lakewood, NJ, USA) through a Conflo II interface (ThermoFinnigan, Breman, Germany). Tin-encapsulated samples were combusted at 1020 °C in a quartz column containing chromium oxide and silvered cobaltous/cobaltic oxide. After combustion, excess oxygen and nitrogen oxides were reduced in a reduction column (reduced copper at 650 °C). N_2_ and CO_2_ were separated on a GC column prior to isotope-ratio mass spectrometry. A series of international reference materials for δ^15^N (IAEA-N-1, IAEA-N-2, and USGS25) and δ^13^C (NBS 22, IAEA-CH-6, and USGS24) were also analysed along with some test batches. Replicate assays of the laboratory standard acetanilide indicated measurement errors of ±0.15‰ for δ^13^C and δ^15^N.

Delta values are expressed relative to international standards: Vienna Pee Dee Belemnite (VPDB) for δ^13^C and Atmospheric Air for δ^15^N.

### 2.4. Statistical Analysis

Locations were clustered by geographical proximity in three coastal areas: the Iberian NW coast (locations 1–2), the Atlantic coast of Andalusia (3), and the Mediterranean coast (4–6) (Figure 1).

We employed Linear Mixed Models (LMMs) to address the statistical non-independence present in the data [33]. These models were constructed with sex and coastal area as fixed categorical predictors, while location (nested within coastal area) was treated as a random factor. The dependent variables considered were δ^13^C and δ^15^N. In cases where it was necessary to ensure data normality and homoscedasticity, a log transformation was applied. Significance testing for each fixed term was conducted using the F-statistic, employing the restricted maximum likelihood (REML) approach. The coefficient of variation was used to visualize the data dispersion in the three coastal areas.

Mean values ± SE are presented throughout the paper. Statistical analyses were performed using IBM SPSS Statistics 29 software. The biplot graph was made using the SIBER and ggplot2 packages in the R programming environment version 4.3.3. 

## 3. Results

The mean δ^13^C values were −16.47 ± 1.05‰ (n = 18) for Kentish plovers breeding on the Mediterranean coast, −16.07 ± 1.11‰ (n = 9) for those breeding on the Atlantic coast of Andalusia, and −14.18 ± 0.27‰ (n = 17) for those breeding on the NW Iberian coast. Regarding the δ^15^N, the mean values for the Mediterranean coast, the Atlantic coast of Andalusia, and the NW Iberian coast were, respectively, 13.57 ± 0.57‰, 14.60 ± 0.61‰, and 14.20 ± 0.19‰. The values for the Mediterranean coast and the Atlantic coast of Andalusia were much more widely dispersed than those corresponding to the NW Iberian coast (Figure 2), with coefficients of variation in δ^13^C and δ^15^N of 8% and 6% for the NW Iberian coast, 20% and 12% for the Atlantic coast of Andalusia, and 27% and 18% for the Mediterranean coast.

The LMMs showed that neither δ^13^C nor δ^15^N concentrations were significantly affected by sex or coastal section (Table 1).

## 4. Discussion

In the Kentish plover adults, the post-breeding moult of primaries is complete and descendent [30]. Temporal moulting patterns in the Iberian Kentish plovers is little-known [27]. In other European, Mediterranean, and north African areas, moult mostly starts in the breeding area between June and mid-July [30,31], with most birds completing the process from mid-August to October [31,32]. In our study, 93.2% of the primaries were collected in April and May, and only three plovers were sampled in June. This collection period minimised the risk of obtaining already moulted feathers, so it is reasonable to assume that the isotopic levels reflected in the study corresponded to the habitats used in the post-reproductive period.

Although the LMMs did not show any significant influence of sex and coastal area on isotopic levels, the coefficients of variation suggest different strategies regarding habitat use in the post-breeding period. Kentish plovers breeding on the Mediterranean coast and the Atlantic coast of Andalusia showed greater variability in isotope values than birds in the NW Iberian Peninsula, both in δ^13^C and δ^15^N. Trophic web carbon and nitrogen values are known to differ between habitats [34,35]. Thus, the size of carbon and nitrogen isotopic niches seems to be related to different individual strategies, such as birds making latitudinal movements in coastal environments and others moving to freshwater wetlands in the post-breeding season, implying that δ^13^C and δ^15^N isotope values are lower than in coastal areas [36,37,38]. The coefficient of variation for the Mediterranean birds reveals very marked habitat changes in plover populations. The high δ^13^C values (−6.59‰ or −9.88‰) may indicate that these birds frequented hypersaline environments [37,39] at the time of feather growth. Conversely, low δ^13^C values (−21.92‰ or −20.95‰) indicate that the birds frequented environments with low salinity [37,39], including inland lakes [40]. Such migratory movements have been reported by other authors [27,40]. By contrast, the results obtained for the northwestern Iberian plover indicate the greater fidelity of this population to a single habitat type, i.e., coastal beaches, with a lower coefficient of variation. In this population, most ringed birds exclusively occupied coastal beaches throughout the annual cycle [41] (unpublished data), although the tracking of some tagged plovers revealed that they use freshwater wetlands in winter (unpublished data). Combining isotopic levels with other techniques, such as tracking birds using telemetry, would be necessary to better identify the areas used by the Iberian population in autumn and winter.

In recent years, beaches have been subjected to strong anthropogenic pressure (urban planning, industrial development, and tourism) [6,42]. The rise in sea level due to climate change is an additional threat [43,44,45]. The most widely accepted projection for the next 100 years is a global sea level rise of 0.63–1.02 m under a scenario of very high greenhouse gas emissions [46]. More specifically, a sea level rise of 0.6–0.8 m is expected in the northwest Iberian Peninsula by 2100 [47]. The inundation of the intertidal zone will result in the conversion of intertidal habitats to subtidal habitats [6] and lead to reductions in habitat availability for many shorebirds. Species that use a wide range of habitats may be better able to cope with these rapid climatic or habitat changes [48,49,50]. By contrast, 90% of the Kentish plovers in the Iberian Peninsula are concentrated on coastal sandy beaches, although a wide variety of habitats are occupied by this species, including salt pans, salt marshes, and endorheic lagoons [2,23,27]. Specifically, in the NW Iberian Peninsula, beaches are the only habitat available for the species [51,52]. In the absence of alternative habitats, the expected rise in sea level due to the effect of climate change will therefore pose a serious problem for the survival of this population, as it will cause a gradual reduction in the availability of suitable habitats for the species. A similar problem has been observed in other coastal populations of this species [53] and of the Snowy plover (*Charadrius nivosus*) [54]. Population decline can be slowed down by implementing effective conservation measures such as increasing survival rates or nest success by implementing actions such as nest protection measures, surveillance, headstarting, or predator control [51,54], as well as the protection and conservation of coastal areas with dune vegetation, for example, by restricting access to sensitive areas, and the restoration of dune environments [28,51]. However, habitat loss can only be compensated for by the creation of alternative habitats.

## 5. Conclusions

This study highlights the heterogeneity in the post-breeding dispersal strategies of Iberian Kentish plovers, with populations in the southern Iberian Peninsula and the Mediterranean showing a wider range of habitats than those in the NW Iberian Peninsula, which are much more dependent on coastal environments and therefore more vulnerable to rising sea levels.

## Figures and Tables

**Figure 1 animals-14-01208-f001:**
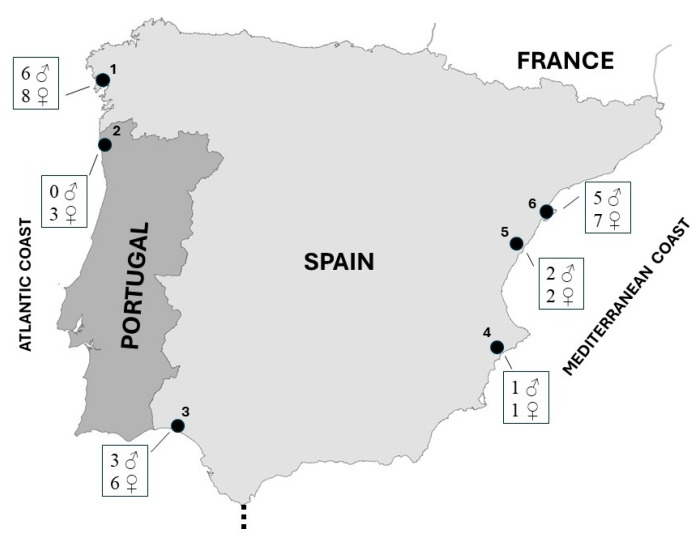
Kentish plover sampling sites used in this study. Locations: 1, Galician beaches (Rostro, Carnota, and Balieiros beaches); 2, Carreço and Ancora beaches; 3; Tinto-Odiel estuary; 4, Laguna de la Mata; 5, Serradal beach; 6, Delta del Ebro. Sample size by sex is shown for each locality.

**Figure 2 animals-14-01208-f002:**
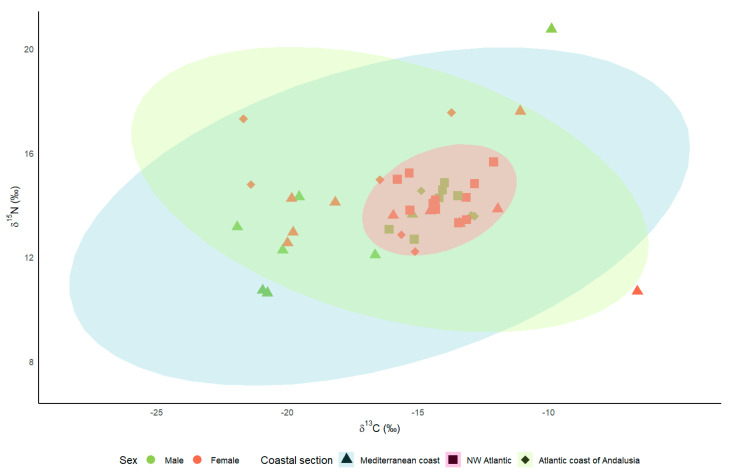
Stable isotope Bayesian ellipses encompassing 95% of δ^13^C and δ^15^N data for Kentish plover by coastal section.

**Table 1 animals-14-01208-t001:** Linear Mixed Models using δ^13^C and δ^15^N in Iberian Kentish plover feathers as response variables, coastal section and sex as fixed categorical predictors, and location (nested within coastal section) as random factor.

Predictors	δ^13^C	δ^15^N
Coastal section	0.325	0.013
Sex	0.612	2.322

## Data Availability

The datasets generated and/or analysed during the current study are available upon request to the corresponding author.

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
