# Peer review of "Use of Stable Isotopes (δ13C and δ15N) to Infer Post-Breeding Dispersal Strategies in Iberian Populations of the Kentish Plover"

_animals, 2024, doi:10.3390/ani14081208_

Round 1
Reviewer 1 Report
Comments and Suggestions for Authors
Manuscript ID: animals-2934691
Title: Use of stable isotopes (δ13C and δ15N) to infer post-breeding dispersal strategies in Iberian populations of the Kentish plover (Charadrius alexandrinus)
Authors: Andrea Gestoso, María Vidal, Jesús Domínguez
Review
I find this manuscript a good example of a short and targeted article. Authors define the aim of the study as “an objective to unravel the post-breeding dispersal strategy in Iberian Kentish plovers by analysis of the isotopic levels of C and N in feathers of breeding adults.” Sample was collected in wide geographic area, analysis done and conclusion reached. Manuscript definitely deserves publication, though I still have two major and several minor comments.
Analysis of SI data
By default, analyses of SI data are conducted using SIBER in R, which produces ellipses in the graphical output. In this case, Figure 2 should be augmented with separate SIBER analyses based on sex and coastal section. Alternatively, these analyses could be represented as isotopic biplots. Either approach will visualize the overlap or differences between males and females, as well as between coastal sections.
Simple summary and Abstract texts
Please acknowledge, that text on the different use of beaches and different dispersals is not based on results of SI analyses.
Minor comments
1. Title and keywords: could you shorten Title, leaving Latin name for keywords only?
2. Use en-dash for ranges, e.g. {4–6], not [4-6]
3. 2.3 perhaps is “Isotopic Analysis”, or “Analysis of Stable Isotopes”?
4. Go through text, there is a space between sign of permille and number in some cases.
5. I think, “-16.07‰±1.11” should be presented as “–16.07±1.11‰”, in all the text.
6. Caption of the Table 1: as there are no differences, part of the caption is not needed.
7. Back Matter is not full, see Template
8. Page ranges in References should use en-dash, DOI are not presented.
Comments on the Quality of English Language
I think language is acceptable
Author Response
Manuscript ID: animals-2934691
Title: Use of stable isotopes (δ13C and δ15N) to infer post-breeding dispersal strategies in Iberian populations of the Kentish plover (Charadrius alexandrinus)
Authors: Andrea Gestoso, María Vidal, Jesús Domínguez
Dear anonymous reviewer,
We address below the response to your comments on the manuscript.
Reviewer comment:
Analysis of SI data: By default, analyses of SI data are conducted using SIBER in R, which produces ellipses in the graphical output. In this case, Figure 2 should be augmented with separate SIBER analyses based on sex and coastal section. Alternatively, these analyses could be represented as isotopic biplots. Either approach will visualize the overlap or differences between males and females, as well as between coastal sections.
Authors answer: Thank you for recommendation. We change the Figure 2 following your suggestion.
Reviewer comment:
Simple summary and Abstract texts: Please acknowledge, that text on the different use of beaches and different dispersals is not based on results of SI analyses.
Authors answer: Ok, we address this issue in the text.
Reviewer minor comments:
- Title and keywords: could you shorten Title, leaving Latin name for keywords only?
Ok, we change it.
- Use en-dash for ranges, e.g. {4–6], not [4-6]
Ok, we change it.
- 3 perhaps is “Isotopic Analysis”, or “Analysis of Stable Isotopes”?
Ok, we correct it for Isotopic Analysis.
- Go through text, there is a space between sign of permille and number in some cases.
Ok, we correct it.
- I think, “-16.07‰±1.11” should be presented as “–16.07±1.11‰”, in all the text.
Yes, we correct it.
- Caption of the Table 1: as there are no differences, part of the caption is not needed.
Ok, we delete it.
- Back Matter is not full, see Template
Ok, we add what’s missing.
- Page ranges in References should use en-dash, DOI are not presented.
Ok, we change it and add the DOIs.
Reviewer 2 Report
Comments and Suggestions for Authors
Comments
- P.2: Salt pans and wetlands are included among alternative areas… are there important man-made wetlands, such as fish farms / fish ponds that could support shorebirds population during fall or spring migration especially when fish harvested? You mentioned only the shores of reservoirs or islands.
Methods:
- P. 3: 44 adult Kentish plovers (17 males and 27 females) … are they sufficient for your study proposal ?
- P 3: the ratio between males and females does not seem balanced. Could you explain if this proportion is good from a statistically point of view ?
- P 3: could be a problem if some birds were in the moulting period when you collected feathers ?
- P 3: for material storage and chemical analyses did you use some protocols or could you cite some previous studies ?
- P 4: Linear mixed models (LMMs) were used… could you or would it have been useful to have used other models ? I think you should try other models if you consider.
Results:
- P4: I think these results should be correlated with those obtain by other classic methods, such as satellite telemetry or even ringing with plastic-colored rings. Such methods are obviously more time-consuming, but can in certain situations give more accurate results Discussion: - P5: at Discussion you were used very frequent suggest or may indicate… I think you should be more precise in the discussions and obviously in the conclusions, if the method and the results allow
Suggestions
1. Have you thought of further using remote sensing to analyze the differences between the foraging and resting areas during post-breeding period ? 2. Could you further use telemetry to know exactly the movement of birds and then to compare to your results based on stable isotopes from feathers ? I suggest to try a more complex study by combining all these methods not just the isotopes Comments on the Quality of English LanguageAuthor Response
Manuscript ID: animals-2934691
Title: Use of stable isotopes (δ13C and δ15N) to infer post-breeding dispersal strategies in Iberian populations of the Kentish plover (Charadrius alexandrinus)
Authors: Andrea Gestoso, María Vidal, Jesús Domínguez
Dear anonymous reviewer,
We address below the response to your comments on the manuscript.
Comments
- P.2: Salt pans and wetlands are included among alternative areas… are there important man-made wetlands, such as fish farms / fish ponds that could support shorebirds population during fall or spring migration especially when fish harvested? You mentioned only the shores of reservoirs or islands.
Kentish plover is a visual foraging species that prey on invertebrates, crustaceans, polychaetes and larvae and adults of insects. It feeds in shores of salt marshes, rice fields, intertidal mudflats, beaches, lagoons, and other inland wetlands.
Methods:
- P. 3: 44 adult Kentish plovers (17 males and 27 females) … are they sufficient for your study proposal?
Indeed, the sample size is not large. It is important to consider that sampling was conducted across a vast study area, involving the search for incubating nests and the capture of birds, which is a complex task in itself. Nevertheless, the sample size is sufficient for robust statistical analysis.
- P 3: the ratio between males and females does not seem balanced. Could you explain if this proportion is good from a statistically point of view?
The sample is unbalanced, yet it still permits reasonably robust statistical analysis. This is attributed to the species' incubation rhythms, as males predominantly engage in incubation during the night, which complicates their capture.
- P 3: could be a problem if some birds were in the moulting period when you collected feathers?
In the Kentish plover adults, the post breeding moult of primaries is complete and descendent. Temporal moulting patterns in the Iberian Kentish plovers is little known. In other European, Mediterranean and north Africa areas, moult mostly start in the breeding area between June and mid-July, with most birds completing the process from mid-August and October. In our study, 93.2% of the primaries were collected in April and May, only 3 plovers were sampled in June. This collection period minimized the risk of obtaining already moulted feathers, so it is reasonable to assume that the isotopic levels reflected in the study corresponded to the habitats used in the post-reproductive period.
- P 3: for material storage and chemical analyses did you use some protocols, or could you cite some previous studies?
We follow the analysis protocol used in other studies, such as the case of: Moon, Y. M., Kim, K., Ki, J., Kim, H. & Yoo, J. C. 2020. Use of stable isotopes (δ2H, δ13C and δ15N) to infer the migratory connectivity of Terek Sandpipers (Xenus cinereus) at stopover sites in the East Asian-Australasian Flyway. Avian Biology Research, 13 (1-2): 10-17.
- P4: Linear mixed models (LMMs) were used… could you or would it have been useful to have used other models? I think you should try other models if you consider.
We consider this multivariate analysis to be appropriate given the characteristics of the sample (dependent variable and predictors).
Results:
- P4: I think these results should be correlated with those obtain by other classic methods, such as satellite telemetry or even ringing with plastic-colored rings. Such methods are obviously more time-consuming, but can in certain situations give more accurate results.
The Kentish Plover is a small sized species with no easy ring reading and recoveries of ringed birds are rare outside your own study area.
Combining isotopic levels with other techniques, such as tracking birds using telemetry, would be necessary to better identify the areas used by the Iberian population in autumn and winter.
Discussion:
- P5: at Discussion you were used very frequent suggest or may indicate… I think you should be more precise in the discussions and obviously in the conclusions, if the method and the results allow
Ok
Suggestions
- Have you thought of further using remote sensing to analyze the differences between the foraging and resting areas during post-breeding period?2. Could you further use telemetry to know exactly the movement of birds and then to compare to your results based on stable isotopes from feathers? I suggest trying a more complex study by combining all these methods not just the isotopes.
Authors answer:
Indeed, we have initiated an ambitious radio-tracking study of individuals at the Iberian level, and among other investigations, we will conduct the analysis combining both methods.
Reviewer 3 Report
Comments and Suggestions for Authors
The authors present a study using stable isotopes to infer post-breeding dispersal strategies in the Kentish Plover Iberian populations. It sheds light on habitat-use strategies in the face of environmental changes. The authors address several questions: How do the Kentish plovers from different regions of the Iberian Peninsula differ in their post-breeding dispersal strategies? What are the main threats to their survival? The language in the text is clear, concise, and technically sound. I found a few glitches and these are mentioned later. Overall, the statistical methods employed are appropriate for the research questions and contribute to the reliability and validity of the study findings. The study provides insights that can aid in protecting Kentish plovers and other migratory bird species in coastal habitats.
However, I am concerned about the sample sizes from the different sites. I appreciate the effort in trapping the birds, but a larger sample size would enhance the statistical power and robustness of the findings. Also, the paper lacks a detailed discussion on the methodology of stable isotope analysis. Further, the paper could benefit from a more extensive discussion comparing its findings with previous studies on Kentish plovers or other shorebird species. This would help contextualize the results and highlight the novelty of the research. I was disappointed that while the paper mentions the threats faced by Kentish plovers, it could delve deeper into the specific conservation implications of the findings. Providing recommendations for conservation actions based on the results would strengthen the practical relevance of the study.
To correct:
Abstract, 1st line – word missing: Beaches are among the ?????? habitats most frequented by migratory birds ……………..
Please write out all numbers below 10.
Author Response
Manuscript ID: animals-2934691
Title: Use of stable isotopes (δ13C and δ15N) to infer post-breeding dispersal strategies in Iberian populations of the Kentish plover (Charadrius alexandrinus)
Authors: Andrea Gestoso, María Vidal, Jesús Domínguez
Dear anonymous reviewer,
We address below the response to your comments on the manuscript.
The authors present a study using stable isotopes to infer post-breeding dispersal strategies in the Kentish Plover Iberian populations. It sheds light on habitat-use strategies in the face of environmental changes. The authors address several questions: How do the Kentish plovers from different regions of the Iberian Peninsula differ in their post-breeding dispersal strategies? What are the main threats to their survival? The language in the text is clear, concise, and technically sound. I found a few glitches and these are mentioned later. Overall, the statistical methods employed are appropriate for the research questions and contribute to the reliability and validity of the study findings. The study provides insights that can aid in protecting Kentish plovers and other migratory bird species in coastal habitats.
However, I am concerned about the sample sizes from the different sites. I appreciate the effort in trapping the birds, but a larger sample size would enhance the statistical power and robustness of the findings.
Authors answer:
Indeed, the sample size is not large. It is important to consider that sampling was conducted across a vast study area, involving the search for incubating nests and the capture of birds, which is a complex task in itself. Nevertheless, the sample size is sufficient for robust statistical analysis.
The sample is unbalanced (males-females), yet it still permits reasonably robust statistical analysis. This is attributed to the species' incubation rhythms, as males predominantly engage in incubation during the night, which complicates their capture.
------------------------------
Also, the paper lacks a detailed discussion on the methodology of stable isotope analysis.
Authors answer: We follow the analysis protocol used in other studies, such as the case of Moon, Y. M., Kim, K., Ki, J., Kim, H. & Yoo, J. C. 2020. Use of stable isotopes (δ2H, δ13C and δ15N) to infer the migratory connectivity of Terek Sandpipers (Xenus cinereus) at stopover sites in the East Asian-Australasian Flyway. Avian Biology Research, 13 (1-2): 10-17.
-------------------------
Further, the paper could benefit from a more extensive discussion comparing its findings with previous studies on Kentish plovers or other shorebird species. This would help contextualize the results and highlight the novelty of the research.
Authors answer: We have already discussed previous studies that support our findings, e.g. references 27, 40, 41, 51-54.
------------------------------
I was disappointed that while the paper mentions the threats faced by Kentish plovers, it could delve deeper into the specific conservation implications of the findings. Providing recommendations for conservation actions based on the results would strengthen the practical relevance of the study.
Authors answer:
In this study we observed great homogeneity in the values recorded for the birds of the northwest of the peninsula, suggesting the use of a single habitat type, the sandy beaches. In the discussion we explained that due to the rise in sea level, the habitat available for the species that inhabit the beaches will be reduced significantly. The specific implication of our findings for conservation is that, in the absence of alternative habitats, the projected rise in sea level due to climate change will therefore pose a serious problem for the survival of this population, as it will lead to a reduction gradual increase in the availability of suitable habitats for the species.
Ok, we include recommendations for conservation actions.
--------------------------
To correct:
Abstract, 1st line – word missing: Beaches are among the ?????? habitats most frequented by migratory birds ……………..
Authors answer: There is no missing word
Please write out all numbers below 10.
Authors answer: Ok
Round 2
Reviewer 2 Report
Comments and Suggestions for Authors
No comment.
I appreciate that the authors have mostly made changes according to the comments and suggestions, and the current form of the paper is an adequate one